# Clinical Guideline on Perioperative Management of Patients with Advanced Chronic Liver Disease

**DOI:** 10.3390/life13010132

**Published:** 2023-01-03

**Authors:** Lidia Canillas, Amalia Pelegrina, Juan Álvarez, Elena Colominas-González, Antonio Salar, Lluís Aguilera, Fernando Burdio, Antonio Montes, Santiago Grau, Luis Grande, José A. Carrión

**Affiliations:** 1Department of Medicine and Life Sciences, Universitat Pompeu Fabra, 08003 Barcelona, Spain; 2Liver Section, Gastroenterology Department, Hospital del Mar, 08003 Barcelona, Spain; 3IMIM (Hospital del Mar Medical Research Institute), 08003 Barcelona, Spain; 4Department of Surgery, Hospital del Mar, 08003 Barcelona, Spain; 5Anesthesia Department, Hospital del Mar, 08003 Barcelona, Spain; 6Pharmacy Department, Hospital del Mar, 08003 Barcelona, Spain; 7Haematology Department, Hospital del Mar, 08003 Barcelona, Spain; 8Department de Medicina, Universitat Autònoma de Barcelona, 08003 Barcelona, Spain

**Keywords:** hepatic, surgery, mortality, risk

## Abstract

(1) Background: Patients with advanced chronic liver disease (ACLD) are living longer with more comorbidities because of improved medical and surgical management. However, patients with ACLD are at increased risk of perioperative morbidity and mortality; (2) Methods: We conducted a comprehensive review of the literature to support a narrative clinical guideline about the assessment of mortality risk and management of perioperative morbidity in patients with ACLD undergoing surgical procedures; (3) Results: Slight data exist to guide the perioperative management of patients with ACLD, and most recommendations are based on case series and expert opinion. The severity of liver dysfunction, portal hypertension, cardiopulmonary and renal comorbidities, and complexity of surgery and type (elective versus emergent) are predictors of perioperative morbidity and mortality. Expert multidisciplinary teams are necessary to evaluate and manage ACLD before, during, and after surgical procedures; (4) Conclusions: This clinical practice document updates the available data and recommendations to optimize the management of patients with advanced chronic liver disease who undergo surgical procedures.

## 1. Introduction

Cirrhosis or advanced chronic liver disease (ACLD) of any etiology is the end-stage and the key risk factor for the development of hepatocellular carcinoma (HCC). ACLD is the third cause of mortality in people aged 45 to 64, and together with HCC is responsible for 3.5% of deaths globally [1]. Frequent etiologies are metabolic-associated fatty liver disease (MAFLD), alcohol liver disease, and viral hepatitis (B and C) [2,3]. The healthy liver parenchyma is replaced by fibrotic tissue and regeneration nodules after a long asymptomatic period with inflammation and fibrosis. The presence of clinically significant portal hypertension (CSPH) is the essential process to develop hepatic decompensation such as ascites or hydrothorax, hepatic encephalopathy (HE), variceal bleeding, as well as complications such as bacterial infections. It is strongly related to a higher mortality risk after surgical procedures that are generally safe in the general population [4,5]. In-hospital mortality rates after non-transplant surgical procedures range from 8.3% to 25%, especially if ACLD patients are decompensated, compared to 1.1% in the general population [6,7]. Moreover, the increase in life expectancy has made individuals require more invasive procedures and a higher incidence of comorbidities such as obesity and cardiopulmonary diseases, which are risk factors for medical and surgical complications independent of the severity of the liver disease. Therefore, the diagnosis of ACLD before surgical procedures is essential because it can allow a better evaluation not only of liver function and CSPH but also of nutritional state, comorbidity, and type of surgery for risk stratification and better management after surgery.

The present document updates the available data and tries to answer the most controversial questions that can be found in our clinical practice regarding the evaluation and management of ACLD patients undergoing surgical procedures.

## 2. Materials and Methods

Current recommendations are based on the European Association for the Study of the Liver (EASL) review for the assessment of non-hepatic surgery risk in cirrhotic patients [6] and the intensive care management of acute-on-chronic liver failure (ACLF) [8], the American Gastroenterology Association (AGA) clinical practice update about perioperative management in cirrhosis [9,10], and the American Association for the Surgery of Trauma (AAST) consensus document on the management of decompensated cirrhosis in the surgical intensive care unit [11].

To date, the clinical guidelines or main documents on the subject focus on some aspects of the perioperative process, such as preoperative evaluation, general postoperative management, or management in a critical care unit. In addition to updating the evidence, we used a new approach to create this clinical practice document. This review aims to answer frequent and controversial questions that we can find throughout the entire perioperative process regarding the evaluation and management of ACLD patients who undertake surgical procedures. The work group was made up of 7 physicians from four different specialties involved in the management of these patients (2 hepatologists, 2 anesthesiologists, 2 surgeons, and 1 hematologist) and 1 clinical pharmacist. A panel of key questions (Table 1) based on our clinical experience was created following the methodology adopted by the “European Association for the Study of the Liver Disease” (EASL) according to the “PICO” structure (P Patient, Population or Problem; I Intervention, Prognostic Factor, or Exposure; C Comparison or Intervention; O Outcome) [12].

Questions were grouped into 5 sections, and at least two experts reviewed each topic: (1) preoperative evaluation and management (LC, AP, JAC), (2) surgical procedures (AP, FB), (3) intraoperative management (JA, LLA) and two sections about postoperative management with (4) general recommendations (EC, AS, JA, JAC); and (5) specific recommendations for hepatic decompensation (LC, JAC). The revision of the topic of each question was carried out according to the Delphi method to achieve the maximum academic consensus [12]. The evidence and recommendations were graded according to the Grading of Recommendations Assessment, Development, and Evaluation (GRADE) system [13]. To simplify, the GRADE levels of evidence were grouped into three categories: A-Meta-analysis and controlled trials, B-Non-randomized trials, cohort and case-control studies, and retrospective case series, and C-Expert opinion. The degree of recommendation was classified as strong (according to quality of evidence, patient outcomes, and costs) and weak (agreeing to variability in preferences and values, or more uncertainty). Each response included the level of evidence (A, B, or C), the degree of recommendation (strong or weak), and the consensus achieved among the experts (from 0/8 to 8/8). Finally, the content was evaluated by three external reviewers, an expert surgeon (LG), a pain physician (AM), and a clinical pharmacist (SG), and they adjusted the final recommendations and the degree of consensus (maximum 11/11) to their recommendations.

## 3. Results

### 3.1. Section 1. Preoperative Evaluation and Management

#### 3.1.1. Pathophysiology


**Q1: What pathophysiological characteristics do patients with ACLD present?**


Portal hypertension is central in the transition from compensated to decompensated liver disease such as ascites or hydrothorax, variceal bleeding, and HE, as well as the increased prevalence of complications such as acute kidney injury (AKI), bacterial infections, and ACLF. Patients with ACLD exhibit an increased intrahepatic resistance to portal blood and therefore in portal pressure, due to distortion of the hepatic microarchitecture (architectural component) and disbalance in vasodilators and vasoconstrictors in the endothelium (dynamic component). In addition, there is an increase in splanchnic blood flow that leads to the development of splenomegaly, hypersplenism, and thrombocytopenia which will affect primary hemostasis [5,7].

The response is the growth of portosystemic collaterals that will further increase the portal flow. Changes in the gut permeability will increase the transfer of bacteria or bacterial products (pathogen-associated molecular patterns; PAMPs) to blood. The passage of ammonium and bacteria products through portosystemic venous collaterals to systemic circulation will produce HE and systemic inflammation [5,7,14].

The splanchnic vasodilatation conduces to systemic vasodilatation with a reduction in the effective arterial blood volume (high compliance for relative blood volume) which manifests with arterial hypotension. As a consequence, neurohumoral vasoconstrictive systems (sympathetic nervous system, renin-angiotensin system, and the non-osmotically stimulated secretion of vasopressin) will be activated to promote sodium and water renal retention in order to increase vascular volume. However, their excessive activation will lead to an excessive plasma volume, sodium-water retention (ascites and hydrothorax), hypervolemic hyponatremia, and eventually kidney dysfunction [5]. The systemic vasodilatation, the hypervolemia, and the stimulation of β1-adrenoreceptor will increase the heart rate and cardiac output leading to a hyperdynamic state [7].

The reduction in normal liver functions appears progressively as the disease evolves. ACLD patients display a reduction in protein synthesis such as albumin and coagulation factors causing hypoalbuminemia and aggravating the free water accumulation, and coagulation disorders. Moreover, malnutrition is frequent in ACLD patients [15].

**Recommendation.** A better understanding of ACLD pathophysiology is necessary to identify signals and mediators of hepatic decompensation or renal dysfunction in front of any invasive procedure.

Level of evidence: A; Recommendation: Strong; Coincidence: 11/11.

#### 3.1.2. Diagnosis of Advanced Chronic Liver Disease


**Q2. How could we diagnose patients with ACLD before surgery?**


Patients with ACLD have non-specific symptoms such as fatigue, stomach pain, anxiety, depression, poor focus and memory, and impaired sleep, which affect individuals’ relationships and ability to work and perform day-to-day tasks. However, not all patients are aware that their symptoms may be related to ACLD. Patients with ACLD have a significant impact on quality of life, with individuals experiencing worse physical and mental health compared with the general population. Patient-reported outcomes are critical to evaluate clinically effective treatments and supportive care. However, more studies are needed to inform patients, include these findings in clinical practice, and influence treatment planning including surgery [16].

Liver biopsy remains the gold standard to quantify the fibrosis stage. A METAVIR fibrosis stage of 4 (F4) identifies cirrhosis [17]. The hepatic venous pressure gradient (HVPG) is the best surrogate of portal hypertension, and a value of 10 mmHg defines CSPH, which is the best predictive variable to identify patients at risk of developing varices and clinical decompensation [18]. Since liver biopsy and HVPG are invasive methods implying risks, the current guidelines recommend that patients with suspicion of ACLD should initially be assessed using non-invasive methods [19].

The simplest non-invasive methods are the serological non-invasive tests (NITs) such as the aspartate aminotransferase (AST) to platelet ratio (APRI) and fibrosis-4 index (FIB-4). Serological NITs cut-offs suggestive of cirrhosis are APRI > 2 and FIB-4 > 3.25 [20,21]. Ultrasonography (US) is a non-invasive and inexpensive radiological technique frequently used as a first-line examination in diagnosing and following up hepatic diseases. The most accurate single sign for the diagnosis of cirrhosis is nodularity of the liver surface and the most commonly associated sign with the presence of portal hypertension is the spleen dimension [22]. The combination of thrombopenia (platelet count < 150 × 10^9^/L) and splenomegaly (spleen diameter > 12 cm) and/or nodular liver surface identify patients with CSPH [23]. However, the most precise radiological technics are elastographic methods that assess liver stiffness. Transient elastography (TE) shows the highest accuracy to identify patients with a highly suggestive ACLD (TE > 15 kPa) and CSPH (TE > 25 kPa) [24,25]. In patients with ACLD, current guidelines recommend a US study every 6 months to identify HCC and upper gastrointestinal endoscopy every 2 years to identify esophageal varices [5].

**Recommendation.** Patients with the suspicion of ACLD should be evaluated by hepatologist to identify CSPH before surgical procedures.

Level of evidence: A; Recommendation: Strong; Coincidence: 11/11.

#### 3.1.3. Surgical Risk Prediction


**Q3. What tools are available to assess the risk of surgical mortality before surgery?**


The risk of postoperative complications and mortality after surgery in ACLD patients is related to the degree of liver dysfunction and portal hypertension, cardiopulmonary and renal comorbidities, type and anatomic site of the surgical procedure, and expertise of the multidisciplinary team (Figure 1). The severity of liver and kidney dysfunction appears to be the most important factor for mortality risk and has been traditionally assessed by the Child-Turcotte-Pugh (CTP) score and the Model for End-Stage Liver Disease (MELD) score. The MELD score includes serum bilirubin, the international normalized ratio (INR), and serum creatinine. Although not initially designed for surgical risk assessment, higher MELD values were associated with higher postoperative mortality (a 1% mortality risk increase per MELD point if MELD < 20 and a 2% mortality risk increase per MELD point if MELD >20) [26]. Nevertheless, CTP or MELD scores do not include comorbidity and surgery-specific variables.

In 2007, Teh et al. evaluated retrospectively short- and long-term mortality in 772 patients with cirrhosis subject to major digestive, orthopedic, and cardiac surgical procedures [27]. Based on their data, the Post-operative Mayo Risk Score (MRS) was designed with the variables that best discriminated mortality risk: age, ASA, MELD, and etiology of cirrhosis (alcoholic/cholestatic vs. viral/others) [27]. However, over the years, there have been significant changes, not only in the higher ages but also in surgical techniques and postoperative management that can lead to overestimation of surgical mortality risk using MRS.

In 2020, Mahmud et al. proposed a new surgical risk scale: the VOCAL-Penn Cirrhosis Surgical Risk Score [28]. The study evaluated factors related to surgical short-term mortality (at 30, 90, and 180 days) in an American retrospective cohort of 3785 patients with cirrhosis who underwent 4712 abdominal (open and laparoscopic), abdominal wall, major orthopedic, vascular, or cardiothoracic surgeries. Patients with ASA V, surgery categories with less than 50 procedures performed, and hepatic surgeries were excluded from the analysis. The VOCAL–Penn score included 9 variables: age, obesity, ACLD etiology (MAFLD vs. others), ASA classification, albumin and bilirubin levels, platelet count, and two surgical variables (emergency and type of surgery). Authors found in the models that platelet count was interchangeable by ascites as a surrogate of portal hypertension, and they maintained platelet count as an objective variable. In addition, the VOCAL–Penn score includes MAFLD as a surrogate of cardiovascular risk. This new score showed an improvement in mortality risk discrimination (AUROC) when compared with preexisting scores (VOCAL–Penn 0.80–0.87 vs. MRS 0.73–0.77, MELD 0.66–0.72, and CTP 0.59–0.68). Recently, the VOCAL–Penn score has been validated in the United States [29], in patients undergoing liver resection [30], and for predicting postoperative decompensation and infection [31].

In MRS and VOCAL–Penn scores, comorbidity is assessed using the ASA scale [27,28]. However, two other comorbidity scales have been validated in patients with cirrhosis: the Charlson Comorbidity Index (CCI) and the Cirrhosis Comorbidity score (CirCom) [32,33]. The CCI was not specifically designed in the 1980s for ACLD patients, and it includes 17 comorbidities. The CirCom is easier to apply, designed for patients with cirrhosis, and provides a better prediction of mortality but was developed on a single-country population cohort with a predominance of alcoholic cirrhosis.

So, there is no single definitive risk stratification system to determine operative risk in all patients with ACLD, and the surgical risk is continuous with no cut-off values for excluding patients from surgical procedures. However, patients with a higher risk of postoperative complications and death (CTP class C or MELD >20) should be evaluated for liver transplantation. The AGA Guideline recommends a preoperative liver transplant evaluation when the predicted postoperative 3-month mortality rate is greater than 15% [9].

**Recommendation.** Multidisciplinary teams have to use validated tools to assess preoperative surgical mortality risk. MRS and VOCAL–Penn scores have online calculators available for public use. However, there are no absolute cut-off values for excluding patients from surgical procedures.

Level of evidence: B; Recommendation: Strong; Coincidence: 11/11.

#### 3.1.4. Preoperative Management


**Q4. How could we intervene to reduce surgical risk before elective procedures?**


The patient and the multidisciplinary team must weigh the potential benefits and risks of surgery collaboratively. There are two types of surgical procedures: (1) lifesaving surgeries, such as cancer surgery, and cardiovascular and emergency interventions, and (2) those that improve patients’ quality of life. Teams have the challenge of deciding whether the procedure can be carried out safely or needs to be delayed until the patient has a better clinical situation (better nutritional status and liver compensation). In some cases, surgery can be delayed until liver transplantation or, sometimes in high-risk patients, is even best avoided [9]. In patients requiring elective surgery, optimization of the patient before surgery if possible should be advocated. In this matter, four key points should be considered (Figure 2). 

Firstly, adequate treatment of underlying ACLD reduces the probability of postoperative decompensation. Thus, in viral hepatitis and autoimmune liver diseases, compliance with the treatment must be ensured, as well as alcohol withdrawal in alcohol liver disease.

Secondly, it is important to ensure proper nutrition and encourage mobility to prevent sarcopenia. Correct nutritional guidelines for ACLD patients include a varied diet of 25 kcal/Kg/day, containing at least 1.2–1.5 g of protein per kg of ideal weight, and frequent meals or snacks throughout the day to avoid fasting greater than 3–4 h. In non-hospitalized obese patients, calorie needs should be adjusted according to the body mass index [34,35].

Thirdly, treatment of ascites and HE based on usual clinical practice [36].

Finally, reduce the risk of bleeding and decompensation due to portal hypertension. In a recent prospective study, an HVPG > 16 mmHg was an independent prognostic factor for postoperative mortality in ACLD patients that underwent extrahepatic surgery [37]. In this regard, a non-invasive evaluation to identify patients with CSPH can facilitate to start non-selective beta-blocker (NSBB) treatment if there is no contraindication. In compensated ACLD patients, carvedilol is the preferred NSBB since it is more effective at reducing HVPG and preventing decompensation with better tolerance. It has a greater portal pressure reducing effect than propranolol, an alternative to carvedilol if it is not well tolerated [38].

In addition, in patients with high-risk varices (large varices or small varices with red dots) with intolerance or contraindication to NSBB, an endoscopic variceal ligation should be recommended to prevent bleeding, although this has no effect on the reduced risk of decompensation [39]. The use of preoperative transjugular intrahepatic portosystemic shunt (TIPS) has been proposed based on case series, reviews of retrospective studies, and retrospective comparative studies, in which the placement of TIPS was safe. In the absence of prospective and comparative studies, the placement of a TIPS before surgery cannot be recommended as a routine practice but it could be considered in patients with HVPG > 16 mmHg that undergo abdominal or esophagal surgery [37,40].

**Recommendation.** The need for prior optimization may determine the best time for elective surgery. Four key points should be considered: adherence to treatment and alcohol withdrawal, improvement in nutritional status and sarcopenia, treatment of hepatic decompensation, and prophylaxis of portal hypertension complications.

Level of evidence: A; Recommendation: Strong; Coincidence: 11/11.

#### 3.1.5. Assessment of Coagulation Disorders


**Q5. How can we evaluate hemostasis and manage coagulation disorders?**


Standard coagulation tests such as prothrombin time (PT), international normalized ratio (INR), and activated partial thromboplastin (aPTT) are not reliable in the prediction of procedural bleeding risk in ACLD patients because these tests are procoagulants-sensitive and insensitive to anticoagulant proteins [9]. According to the 7th International Coagulation in Liver Disease Conference, the PT/INR and the aPTT should be avoided if we have to assess the risk of bleeding or guide blood product transfusion in patients with ACLD. Similarly, the platelet count is not a good indicator of the aggregation capacity because of the increase in the von Willebrand expression [41].

Viscoelastic tests (VETs), thromboelastography (TEG) and rotational thromboelastometry (ROTEM), are methods of testing the efficiency of blood coagulation, and its point-of-care use is increasing in surgery, emergency departments, and intensive care units [42]. Two recent randomized controlled studies (RCTs) showed that most ACLD patients have a normal range of TEG values [42,43]. Current evidence suggests that the use of VET decreases the prophylactical use of blood products in patients undergoing invasive procedures, without a bleeding increase [44]. However, the administration of prophylactic blood products in patients undergoing invasive procedures, especially if they are at low risk of bleeding, is under debate. During liver transplantation, VETs are an important part of the transfusion-free liver transplant. They provide well-correlated parameters with the fibrinogen level (except in severe hypofibrinogenemia), with an intraoperative platelet count and with hypercoagulable states, allowing more dynamic decision-making through a point-of-care system. The use of these tests and transfusion protocols made it possible to reduce the infusion of blood products and the costs in liver transplantation [44]. Usually, these protocols tend to give more importance to the use of fibrinogen concentrate and cryoprecipitate [9]. Moreover, VETs are also efficient in guiding treatment during acute gastrointestinal bleeding episodes in ACLD patients and, maybe in the future, could be useful to guide the anticoagulant therapy [44]. Nevertheless, more studies are needed to assess the normal parameters of these tests and the cut-offs as transfusion triggers in the perioperative period in ACLD patients.

Prophylactic fresh frozen plasma (FFP) transfusion should not be performed, even in patients with highly abnormal PT/INR levels because it can significantly increase portal pressure, risk of bleeding, and side effects on lungs [9,45]. Prothrombin complex concentrate (PCC), fibrinogen concentrates, or cryoprecipitates (containing fibrinogen, von Willebrand Factor, and Factor VIII) are preferred because they balance the hypofibrinogenemia with less volume overload. The use of antifibrinolytic drugs, such as tranexamic acid, is also discouraged. Severe thrombocytopenia is related to periprocedural bleeding events, but there is no evidence that prophylactic platelet transfusion improves the hemostatic potential [41]. In patients with a severe coagulopathy (platelet count < 50,000/μL or fibrinogen level < 100 mg/dL), undergoing high-risk procedures without the possibility of local hemostasis, the prophylactic administration of a platelet or fibrinogen concentrate may be considered (Figure 3) [46,47]. Thrombopoietin receptor (TPO-R) agonists, lusutrombopag or avatrombopag, are approved for regulatory agencies in ACLD patients undergoing invasive procedures with severe thrombocytopenia (platelet count < 50,000/uL). Again, they are not routinely recommended, and their administration can be regarded in elective high-risk surgeries (as patients need to complete a 5- to 7-day course of treatment before the procedure) [46,48]. Moreover, general considerations have been described as beneficial for periprocedural management of hemostasis in ACLD patients as (1) optimize anemia by treating iron, folic acid, and vitamin B6 and B12 deficiencies; (2) treat infections; (3) improve renal function; and (4) avoid acidosis.

In ACLD patients with active bleeding, a fibrinogen level >120 mg/dL is mandatory [48]. Prothrombin concentrates complexes (Coagulation Factors II, VII, IX, X) involve less volume than the FFP. Desmopressin (DDAVP) improves platelet function in patients with renal failure. Nevertheless, some recent randomized control trials found no benefit in its use for bleeding esophageal varices, and the recommendation is to restrict its use to ACLD patients with concomitant kidney disease [45]. Due to the thrombotic risk of tranexamic acid, its use is only recommended when the hyperfibrinolysis status has been identified.

**Recommendation.** Prophylactic transfusion protocols guided by PT/INR are not recommended. Routine platelet transfusion is not advised. Viscoelastic testing (thromboelastography and rotational thromboelastometry) allows a more accurate assessment of coagulation and should be considered to guide the transfusion decision before and during surgery.

Level of evidence: B; Recommendation: Strong; Coincidence: 10/11.

### 3.2. Section 2. Surgical Procedures

#### 3.2.1. General Recommendations


**Q6. What recommendations should be considered for surgical procedures?**


The risk of morbidity and mortality of elective surgeries in ACLD patients varies depending on the type of surgery [49]. Higher mortality risk has been observed in ACLD patients in different surgery types [50]. Among types of surgery, hepatobiliary surgery is perceived to have the highest risk of liver-related mortality and morbidity. Portal hypertension is a significant risk factor in thoracic and esophagogastric surgery [9]. Elective colorectal surgeries are at higher risk of morbidity and mortality, and higher MELD-Na values have been related to an anastomotic leak in rectal resections [51].

The laparoscopic approach was linked to a decreased rate in postoperative complications but acknowledged possible selection bias [52]. Several case series have proposed that, compared with more invasive approaches, laparoscopic and other minimally invasive surgical techniques (the robotic approach) may lead to favorable outcomes in ACLD patients for benign diseases, HCC, or colorectal liver metastases as it reduces the incidence of post-operative ascites, liver failure, and morbidity [53]. However, in lesions requiring biliary reconstruction, a minimally invasive approach is a challenging option, in which the robotic approach could represent a useful support [54].

**Recommendation.** ACLD patients have higher risk in all types of surgery, but emergency, hepatobiliary, esophagogastric, and thoracic surgeries are of special risk in patients with CSPH. Laparoscopic and robotic approaches should be preferred to reduce postoperative complications.

Level of evidence: A; Recommendation: Strong; Coincidence: 11/11.

#### 3.2.2. Elective Surgery


**Q7. What considerations should be taken into account for elective surgery?**


The presence of ascites predisposes to hernias in the abdominal wall. Incarcerated hernias are the most common indication for emergency surgery in ACLD and result in higher mortality and morbidity as compared with elective hernia repair [55]. On one hand, elective abdominal hernia surgery should be avoided in ACLD patients with ascites unless the ascites are controlled medically because of the risk of dehiscence and peritonitis. On the other hand, abdominal hernia strangulation with bowel ischemia or gangrene is an emergency situation. An intensive diuretic treatment before and after the intervention and a large-volume paracentesis are recommended to reduce ascites’ recurrence and postoperative complications [9].

Hepatic resection for HCC is considered in patients with Barcelona Clinic Liver Cancer stage 0 or A HCC [56]. The main concern with liver resection is the risk of liver failure, especially in patients with ACLD. Its incidence has been reported in around 8–12% of hepatectomies. Several factors are associated with its appearance, not only variables of the patient, the surgical technique, or the postoperative period but also, most importantly, the liver function [57]. Thus, hepatic resection should be performed in ACLD patients without CSPH (HVPG < 10 mmHg) to avoid clinical decompensation and mortality after surgery [58]. Liver stiffness (TE < 20) and MELD (MELD < 9) scores are good surrogate markers and can be used to select patients for hepatic resection because of their excellent prognosis [59,60,61]. Compared to open resection, laparoscopic liver resection was proved to result in less blood loss, a shorter length of hospital stay, and a lower complication rate [62].

The increasing prevalence of obesity and diabetes has led to an increase in MAFLD ACLD [63]. Currently approved endoscopic bariatric therapies include the intragastric balloon, the percutaneous gastric aspiration system, and the sleeve gastrectomy. Endoscopic bariatric therapies may have lower risks compared with surgical approaches, although direct comparative studies and long-term efficacy data are currently lacking. Laparoscopic sleeve gastrectomy is most likely the optimal bariatric surgical procedure in patients with ACLD due to preservation of endoscopic access to the biliary tree, gradual weight loss, and absence of malabsorption, compared with gastric bypass. Neither endoscopic bariatric therapies nor bariatric surgical procedures should be performed in patients with CSPH. In decompensated liver disease, the only acceptable option at present is bariatric surgery at the moment or after a liver transplant [9].

**Recommendation.** Elective surgeries must be avoided in decompensated patients. Good liver function (MELD score < 9) is recommended before liver resection. Presence of CSPH should be excluded before liver resection and bariatric surgery.

Level of evidence: A; Recommendation: Strong; Coincidence: 11/11.

#### 3.2.3. Emergent Surgery


**Q8. What concerns should be considered for emergent surgery?**


Frequent emergency surgeries are cholecystectomy for cholecystitis, hernia repair for an incarcerated hernia, or colorectal surgery for complicated diverticular disease or colorectal cancer. Emergency interventions for any indication carry a higher risk of mortality and worse outcomes than elective procedures [64]. The risk of postoperative mortality is 4 to 10 times higher, and complications are 5 to 7 times more frequent [10]. This is because the clinical conditions of patients requiring emergency interventions are usually more severe, and there is not time enough to optimize the patient’s situation before the intervention [65].

Incarcerated hernias are the most common indication for emergent surgery in ACLD and result in higher mortality and morbidity as compared with elective hernia repair [55].

The diagnosis of cholecystitis can be challenging because in ACLD patients the gallbladder wall can appear thickened on imaging due to fibrosis and ascites. When the diagnosis of cholecystitis is certain, the patient should be referred to a center with experience in its management in patients with ACLD. In general, cholecystectomy must be avoided in CTP class C patients or those with refractory ascites because of its significant risk of complications. These patients are also generally not candidates for percutaneous cholecystostomy or US-guided gallbladder drainage (unless they are critically ill) due to the presence of ascites, collateral circulation, and the high risk of infections [9,66,67].

**Recommendation.** Emergent surgeries have a higher risk of complications and should be performed in centers with experience in ACLD patients. The best therapeutic option and alternatives to surgical intervention must be assessed by a multidisciplinary team.

Level of evidence: B; Recommendation: Strong; Coincidence: 11/11.

### 3.3. Section 3. Intraoperative Management

#### 3.3.1. Anaesthetic Technique


**Q9. Is the anesthetic technique different?**


Studies in healthy volunteers indicate that liver flow decreases by 35–42% after induction of general anesthesia. Therefore, the goal of anesthesia should be aimed at maintaining adequate liver flow [68]. In ACLD patients, we can find changes in hepatic flow with the creation of arterio-venous fistulas, reduction in hepatocyte function with a decrease in proteins synthesis, and destruction of the bile canaliculi with a decrease in the ability to excrete metabolites through the bile [69]. Moreover, animal models demonstrated that extrahepatic surgery can increase portal pressure [70].

The halogenated drugs are fundamentally absorbed and excreted through the respiratory system and that makes them ideal for ACLD patients. Nevertheless, due to the lipophilic properties, they require a certain degree of hepatic metabolism [71]. The trifluoroacetyl chloride, one of its metabolites, can cause immune-mediated hepatotoxicity.

Sevoflurane has registered the fewest complications. Isoflurane has the lowest degree of hepatic biodegradation (1.7%), and desflurane has also not shown liver damage [72]. In addition, both desflurane and isoflurane have shown little involvement regarding the hepatic flow.

Propofol is a widely used hypnotic drug for induction and maintenance of anesthesia, and there is a clear recommendation for its priority use in ACLD patients undergoing endoscopy [73]. Propofol pharmacokinetics is dependent on binding to proteins, and it is essentially metabolized by the liver. Recovery times are slightly longer in ACLD patients, and the plasma awakening concentration is comparable to the general population (around 1 mcg/mL) because it has a significant extrahepatic metabolism in the lung and kidney [74].

The metabolism of benzodiazepines occurs basically in the liver. Those metabolized by oxidation such as diazepam, clonazepam, and midazolam experience a prolongation of their effect in ACLD patients. In contrast, those metabolized by conjugation such as lorazepam, oxazepam, and temazepam show half-lives minimally affected by liver disease, and they do not present active metabolites. Oxazepam and temazepam can be used as a preoperative anxiolytics, while lorazepam can be used for surgical sedation in countries that have the intravenous presentation.

Succinylcholine is a depolarizing neuromuscular blocking agent mostly hydrolyzed by pseudocholinesterase, so patients with severe liver disease may present a reduction in the activity of these enzymes with a prolongation of the neuromuscular blockade. Vecuronium has also a longer elimination half-life (58 to 84 min) due to delayed elimination and biliary excretion [75]. The rocuronium effect onset remains unchanged and makes it valid for rapid sequence inductions, but the recovery time from the neuromuscular blockade is 50% longer in ACLD patients because of its biliary excretion and the high volume of distribution [76]. Sugammadex is an oligosaccharide that is administered with rocuronium to block its molecule and reverse its effect. Its use is safe and effective also in patients with liver dysfunction. Atracurium, cis-atracurium, and mivacurium are metabolized by the Hofmann degradation in peripheral circulation and tissues without liver involvement. Only in the case of mivacurium and advanced liver disease, the onset and recovery can be delayed by the higher volume of distribution.

Phenyl pyridines (fentanyl, sufentanil, and remifentanil) are used during the anesthesia, and their metabolism is not affected in ACLD patients but can cause HE [77]. Of them, remifentanil metabolization occurs through the plasma esterase function exclusively, making it safe even in severely advanced liver disease. In Table 2, we summarize the characteristics of the main drugs used in anesthesia in patients with ACLD.

**Recommendation.** Sevoflurane is the halogenated anesthetic with the fewest complications in ACLD patients. Propofol is currently a widely used hypnotic drug for the induction and maintenance of anesthesia. Lorazepam, oxazepam, and temazepam are benzodiazepines minimally affected by liver disease.

Level of evidence: B; Recommendation: Strong; Coincidence: 11/11.

#### 3.3.2. Loco-Regional Anaesthesia


**Q10. Is regional anesthesia recommended?**


Regional anesthesia should be the primary choice for ACLD patients, if possible. Local anesthetics directly act on the spinal cord or peripheral nerves, unaffected by pharmacokinetics’ changes due to liver disease. Moreover, its use reduces the increased risk of aspiration with general anesthesia in patients with ascites.

Coagulopathy represents a great limitation for neuraxial anesthesia (epidural or subarachnoid) because of the catastrophic result of a spinal hematoma. However, when possible, its use effectively reduces surgery-induced stress responses (catecholamines and corticosteroids). Extrapolating from a consensus on obstetric anesthesia, thrombopenia with a platelet count >70,000/mm^3^ could be considered safe [78]. The use of VETs (TEG or ROTEM) could identify ACLD patients without hypocoagulability who could be candidates for these techniques, but more studies have to be conducted to support this recommendation [79]. Other low-risk regional techniques, such as plexus, interfacial plane, or paravertebral blocks have more consensus for their use in ACLD patients as they do not have the risk of central nervous hematoma.

When regional anesthesia is not possible as a single anesthetic technique, it is then recommended as part of the multimodal approach during general anesthesia, resulting in a very important enhanced recovery. Pain control with regional anesthesia can reduce the risk of HE, the risk of respiratory depression associated with opioids, the thromboembolic complications, and can improve the splanchnic perfusion with an early recovery of intestinal motility after abdominal surgery [80]. Moreover, as sympathetic activity has been related to liver damage, blocking it with regional anesthesia during surgery could reduce this deleterious effect [81].

**Recommendation.** Regional anesthesia should be the choice if possible. When general anesthesia cannot be avoided, a regional technique could be recommended as part of the multimodal approach to enhance post-surgery recovery.

Level of evidence: B; Recommendation: Strong; Coincidence: 11/11.

#### 3.3.3. Cardiovascular Assessment and Intraoperative Monitoring


**Q11. Is cardiovascular monitoring necessary during surgery?**


Patients with ACLD show an increased cardiac output and peripheral vasodilatation and decreased vascular resistance, systolic dysfunction, and inappropriate response to surgical stress. Around 40–50% of patients express some degree of cardiomyopathy characterized by systolic and diastolic dysfunction and electrophysiological abnormalities, such as QT interval prolongation. Under that condition, the increase in cardiac output cannot compensate for the reduction in arterial pressure and compromises tissue oxygenation, driving multi-organ failure [82]. In addition, they present activation of the renin-angiotensin-aldosterone system that leads to increased sodium and water retention, although the central blood volume remains low. Therefore, cardiovascular monitoring is crucial in the intraoperative period.

Central venous pressure, as a static parameter, is not a good indicator of actual blood volume, especially in the presence of ascites, diastolic dysfunction, and hypoalbuminemia [11,83]. Pulmonary artery catheter insertion is the gold standard for the measurement of preload, afterload, and cardiac output; however, it is an invasive technique associated with a high rate of serious adverse effects and is rarely used in the operating room.

Currently, various noninvasive and minimally invasive measurement systems have been introduced in intraoperative management. Mean arterial pressure (MAP) can be monitored using both an intra-arterial catheter and a non-invasive monitor. A MAP target of 60–65 mmHg in ACLD patients grants organ perfusion [11]. Intra-arterial catheters can be used to quantify pulse pressure variation (PPV) or stroke volume variation (SSV), which are dynamic preload parameters based on the heart-lung interaction, widely used in surgical and critical patients. However, these parameters show some general limitations (sinus rhythm under controlled mechanical ventilation combined with conservative tidal volume settings); the ascites can alter aortic compliance (a determinant of PPV), and the hyperdynamic state and the low systemic vascular resistance can alter their reliability [84]. Transthoracic echocardiography (TTE) allows hemodynamic assessment in a non-invasive way using qualitative parameters (gross appearance, wall motion, and estimation of ejection fraction) and quantitative parameters (cardiac output, left ventricular end-diastolic area, stroke volume variation, change in velocity time integral and dynamic inferior cava diameter). However, the acquisition of the planes necessary for interpretation is limited by some factors such as the experience of the examiner, the position in the operating room, the presence of mechanical ventilation, and ascites. There are also limitations when pulmonary hypertension and cirrhotic cardiomyopathy coexist. The transesophageal alternative is safe in patients with grade 1 or 2 esophageal varices and no recent history of bleeding [84].

Other analytical parameters can provide information about tissue oxygenation as a marker of hemodynamic adequacy. However, again, it must be interpreted carefully in ACLD patients. A low central venous oxygen saturation (ScVO2) indicates an insufficient cardiac output for tissue oxygenation. However, in patients suffering arteriovenous fistulas, it can remain elevated due to the hyperdynamic state. Elevated lactate may mean anaerobic metabolism and also poor tissue perfusion. On the contrary, in ACLD, it may be elevated in a context of hepatic clearance deficiency.

**Recommendation.** Cardiovascular monitoring is crucial in the intraoperative period to avoid hypoperfusion and guide volume resuscitation and vasoconstrictive therapy. It can be performed with different techniques that have their pros and limitations. Static measures of cardiac preload (central venous and pulmonary capillary wedge pressure) are poor indicators of volume status and fluid responsiveness but can be useful to check the effect of fluid administration. Dynamic measures (PPV, SSV, TTE) are preferred.

Level of evidence: A; Recommendation: Strong; Coincidence: 11/11.

#### 3.3.4. Fluids and Vasopressors during Surgery


**Q12. What is the best fluid replacement strategy and vasopressor during surgery?**


Crystalloid-based fluid replacement may be detrimental, as it worsens edema and ascites, showing little effect on intravascular volume. For this reason, the fluid replacement must be goal-directed, according to the monitoring techniques mentioned previously. If the cardiovascular assessment indicates fluid-responsive hypovolemia, balanced solutions should be used, such as Plasmalyte, with a lower content of sodium and chloride and less risk to cause kidney failure than normal saline [11]. Hydroxyethyl starch should not be used due to its potential nephrotoxicity.

In some scenarios, albumin solutions must be the preferent choice. The abrupt release of intra-abdominal pressure during surgery in patients with ascites can lead to circulatory dysfunction, similar to what occurs in large-volume paracentesis. Then, 8 g/L of albumin per liter of ascites liquid should be administered in order to avoid AKI, hyponatremia, or HE [85]. Albumin replacement improves effective blood volume, compensating for hypoalbuminemia, and increasing the intrinsic antioxidant and detoxifying capacity related to this protein [86]. Those properties make albumin solutions essential in the treatment of spontaneous bacterial peritonitis (SBP), hepatorenal syndrome, and refractory ascites.

After a negative fluid challenge that leads to a worsening of arterial pressure or cardiac output, vasoactive drug therapy represents the starting point of the treatment. The vasoactive drug of choice during surgery in ACLD patients is norepinephrine [11,85]. Patients in chronic treatment with beta-blockers have an impaired cardiac response under stress conditions, so in order to avoid hypoperfusion, early use of norepinephrine is mandatory [79].

**Recommendation.** It is essential to assess volume status and fluid responsiveness during fluid therapy in order to avoid over-resuscitation related to ascites, edema, hyponatremia, and portal hypertension worsening. As a general rule, a fluids restriction attitude is well accepted. When losses must be compensated during surgery, balanced crystalloids should be used as first-line treatment, but albumin solutions can offer advantages in some scenarios. Early use of norepinephrine to avoid tissular hypoperfusion should be considered.

Level of evidence: A; Recommendation: Strong; Coincidence: 11/11.

### 3.4. Section 4. General Recomendations in Postoperative

#### 3.4.1. Transfusion Strategy in Acute Bleeding after Surgery


**Q13. Is a liberal or restrictive transfusion strategy recommended in acute bleeding?**


In ACLD patients with acute hemorrhage, a liberal red blood cell (RBC) transfusion strategy was shown to increase portal pressures, which can directly mediate rebleeding. A systematic review/meta-analysis that included five RCTs comparing restrictive versus liberal RBC transfusion reported that a restrictive policy was associated with a significant overall reduction in mortality and rebleeding without a difference in the risk of ischemic events [87]. Therefore, in hemodynamically stable ACLD patients, the European Society of Gastrointestinal Endoscopy (ESGE) recommends a restrictive RBC transfusion strategy, with a hemoglobin threshold of ≤ 70 g/L, if no history of cardiovascular disease is present or ≤ 80 g/L for those with acute or chronic cardiovascular disease history [88]. Similarly, the FFP transfusion is associated with significantly increased mortality, failure to control bleeding, and a longer hospital stay [89]. Lower volume factor replacements such as PCC appear to be more effective than FFP in decreasing INR values [90]. Limited data are available in patients with thrombocytopenia on the requirement for platelet transfusion in active bleeding. However, the use of VET can guide the blood product transfusion strategy recently demonstrated a significant reduction in blood product transfusions and the risk of rebleeding [91].

**Recommendation.** In acute bleeding, a restrictive transfusion strategy is recommended with a hemoglobin threshold ≤ 70 g/L. However, we do not recommend protocol transfusions to an INR/platelet target. The use of VET should be desirable to avoid needless transfusions and volume overload.

Level of evidence: A; Recommendation: Strong; Coincidence: 11/11.

#### 3.4.2. Venous Thromboembolism Prophylaxis


**Q14. Is venous thromboembolism prophylaxis recommended after surgery?**


Patients with ACLD are at risk for thrombosis as well as bleeding. Reductions in liver-derived procoagulants are offset by reductions in anticoagulants as well as increases in endothelial-derived procoagulants, leading to a fragile net hypercoagulable state. As a consequence, the development of deep vein thrombosis (DVP) is 50–70% higher in patients with ACLD. Moreover, portal hypertension produces a reduction in the portal vein flux increasing the risk of portal vein thrombosis [11]. The IMPROVE score (≥4 points) can be used to predict DVP risk in ACLD patients [46]. Studies of venous thromboembolism prophylaxis in ACLD patients are retrospective and included heterogeneous cohorts, so they showed conflicting results on efficacy [46]. Besides these limitations, prophylactic anticoagulation does not increase bleeding events if we use low-molecular-weight heparin (LMWH) and exclude severe renal failure (creatinine clearance < 15 mL/min) [11,46].

**Recommendation.** In high-risk patients without hypocoagulability in VET, venous thromboembolism prophylaxis can be beneficial. The use of LMWH is reasonable after excluding severe renal failure.

Level of evidence: C; Recommendation: Weak; Coincidence: 7/11.

#### 3.4.3. Pain Control


**Q15. What is the treatment of postoperative pain?**


In patients with ACLD, liver dysfunction leads to alteration in drug pharmacokinetics and metabolism thus increasing the risk of toxicity [92]. Major surgery is often followed by moderate to severe pain, so postoperative pain management is a major concern in patients undergoing surgery. General considerations and dose recommendations regarding the use of the most common analgesics in patients with ACLD are described in Table 3. The WHO analgesic ladder has been applied in the management of acute and chronic cancer and non-cancer painful conditions [93]. A simplified algorithm of recommended analgesia in patients with ACLD is depicted in Figure 4.

Although a common misconception, the short-term use of a low dose (2–3 g/day) of acetaminophen is safe in patients with ACLD [92,94,95,96]. In contrast, NSAIDs should be avoided in all ACLD patients, as they can precipitate AKI and gastrointestinal bleeding due to prostaglandin inhibition and increased bioavailability [94,97]. Metamizole (also known as dipyrone) is not a classic COX inhibitor, but recent studies reported an increased risk of AKI [98,99].

**Table 3 life-13-00132-t003:** Recommendations for use of analgesics in patients with advanced chronic liver disease.

Analgesic Drug	General Considerations	Dosing Recommendation
**Acetaminophen/** **paracetamol**	Safe [94,100]	Not exceed the daily dose of 2–3 g [94]
**NSAID** ^1^	Avoid use [94]	Avoid use [94]
**Metamizole/** **dipyrone**	Avoid use [98]	Avoid use [98]
**Codeine**	Avoid use	Avoid use [94]
**Tramadol**	Use with careful monitoring in patients taking selective serotonin reuptake inhibitors or tricyclic antidepressant. Avoid in patients with seizure history [77,94]	Start with 50 mg/day. Maximum 200 mg/d [101]
**Fentanyl**	Use with caution in patients with moderate liver disease [77,100]	Start with half of usual dose [101]
**Morphine**	Use with caution Avoid in renal disease Avoid extended release formulations [94]	Start with half of usual dose [101]
**Oxycodone**	Use with caution Avoid extended release formulations [94]	Start with a quarter of usual dose [101]
**Meperidine**	Avoid use [94]	Avoid use [94]

^1^ NSAIDs: nonsteroidal anti-inflammatory drugs.

Opioids can be given in patients with ACLD under close surveillance with cautiously increasing dosage due to the risk of constipation and triggering HE [101]. In general, because of a decreased drug clearance in ACLD patients, immediate release formulations are preferred over extended-release formulations, and extended dosing intervals should be prescribed [94]. Regarding weak opioids, as tramadol requires conversion to O-desmethyltramadol by hepatic oxidation, the analgesic effects may be unpredictable [77,102]. However, the risk of respiratory depression, which is the most fatal complication of these drugs, appears less with oral tramadol than with morphine. So, tramadol use in ACLD patients may be safe [94,101]. Codeine has limited conversion to the active metabolite and should be avoided due to diminished analgesic properties [94,100]. Regarding strong opioids, fentanyl could be safe in patients with modest hepatic dysfunction as the pharmacokinetics of fentanyl were unchanged when compared with healthy subjects [77,100,103]. However, it is not known if the metabolism of fentanyl is affected in patients with severe hepatic dysfunction. Clearance of morphine and oxycodone is delayed in ACLD patients so they should be used at reduced doses and prolonged intervals of administration to avoid accumulation [94]. Meperidine has unpredictable analgesic effects and an increased risk of toxicity in patients with ACLD so it should be avoided [92,94].

**Recommendation.** Acetaminophen is the preferred analgesic in patients with advanced chronic liver disease. NSAIDs and metamizole should be avoided because they can increase the risk of AKI and gastrointestinal bleeding. Opioids should be used with caution and at reduced doses.

Level of evidence: B; Recommendation: Strong; Coincidence: 11/11.

#### 3.4.4. Nutrition


**Q16. What nutritional recommendations should be made after surgery?**


The liver is involved in glucose, lipid metabolism, and energy homeostasis. Malnutrition is common in ACLD, affecting between 20 and 50% of patients and reaching more than 60% with decompensated cirrhosis. Protein-calorie malnutrition is a prognostic indicator of mortality among hospitalized patients with CSPH [104]. A worsening of nutrition status is correlated with the severity of liver function [11,15]. The pathogenesis is multifactorial with contributions from inadequate intake, impaired digestion and absorption, and altered lipid, protein, and carbohydrate metabolism. The presence of ascites, gastrointestinal bleeding, portosystemic shunting, and bacterial overgrowth may also have an impact on the intake and uptake of macronutrients [34].

Hypoglycemia can occur in patients with decompensated cirrhosis due to depletion in hepatic glycogen stores, impaired gluconeogenesis due to hepatocyte loss, and hyperinsulinemia. Continuous infusion of 10–20% dextrose can mitigate hypoglycemia, although care should be taken with volume overload. Therefore, 30% or 50% dextrose boluses may be used, and more frequent glucose checks (every 2 h) may be beneficial in the case of hypoglycemia [11].

As nutritional reserves have prognostic implications, nutrition and micronutrient supplementation must be started early after surgery (Figure 3). In patients with esophageal varices undertaking extra abdominal surgeries, the insertion of a nasogastric tube for enteral feeding is associated with a low risk of bleeding, and it is not contraindicated [8]. Enteral nutrition (EN) is preferred rather than parenteral feeding as it reduces infectious complications by preserving the intestinal mucosal barrier and healthy microbiota. In addition, shorter intensive care unit (ICU) stay and lower paralytic ileus incidence have been reported with EN [105]. Therefore, parenteral nutrition should be reserved as second-line in cases where EN is contraindicated (gastric residuals > 500 mL, active gastrointestinal bleeding, intestinal ischemia, or abdominal compartment syndrome) or when nutritional requirements alone are not met [11,105]. The goals of nutritional support entail the provision of at least 35 kcal/kg/day and 1.2–1.5 g of protein/kg ideal body weight/day in non-obese and non-critically ill patients. In non-hospitalized obese patients, calorie needs should be adjusted according to the body mass index. In critically ill patients, an increase in protein intake of up to 2 g/kg is recommended [8,34]. There are no current recommendations for the routine use of specific enteral feed formulas or protein restriction, neither in the presence of HE [8,34]. Although it was proposed that branched-chain amino-acids could be beneficial in cirrhosis or HCC, their recommendation in these pathologies is contradictory and, currently, cannot be extended to this population [106].

**Recommendation.** Nutrition and micronutrient supplementation should be started early after surgery, preferably using the enteral route (orally or by placing a nasogastric tube). Before starting nutrition, continuous infusion of 10–20% dextrose may be used to avoid hypoglycemia.

Level of evidence: B; Recommendation: Strong; Coincidence: 11/11.

### 3.5. Section 5. Postoperative Management of Hepatic Decompensation and ACLD Complications

#### 3.5.1. Acute Kidney Injury


**Q17. How should we manage an acute kidney injury after surgery?**


Acute kidney injury (AKI) is a decrease in glomerular kidney function (GFR). The incidence of AKI ranges from 20 to 50% in ACLD patients and increases their morbidity and mortality. Moreover, chronic kidney disease is increasing in ACLD patients as there has been a rise in the prevalence and incidence of type II diabetes mellitus (DM) and obesity in patients with MAFLD. ACLD patients can have all the AKI types (prerenal, intrarenal, and post-renal). Additionally, they can have hepatorenal syndrome (HRS), a type of renal dysfunction resulting from the systemic hemodynamic effects of portal hypertension. Therefore, AKI in ACLD can be separated into non-HRS-AKI and HRS-AKI. Kidney Disease Improving Global Outcomes (KDIGO) provide the most recent consensus definition for AKI; it was updated in 2012, and, in 2015, The International Club of Ascites revised the sCr level of 1.5 mg/dL to differentiate between stage 1-A and stage 1-B [107].

It is necessary to investigate early the cause of AKI to treat it and prevent the progression of the disease. The treatment of AKI in ACLD patients includes a series of general measures: (1) detection and discontinuation of factors contributing to AKI as nephrotoxic medications (NSAIDs and metamizole) [99]; (2) if dehydration, diuretics, and lactulose should be discontinued; (3) patients should have screening for infections and initiate antibiotics promptly if there is suspicion; and (4) volume expansion, when necessary, must be performed in relation to the cause and severity of the loss of volume. In case of gastrointestinal bleeding, transfusion is indicated when Hb < 7 g/dL with an aim of Hb 7–9 g/dL; if the patient requires large-volume paracentesis, 8 g of albumin should be administered per liter of fluid removed (independently of the total volume removed), and patients with AKI stage 2 or 3 should receive volume expansion with albumin 1 g/kg/24 h for 48 h.

Most cases of pre-renal AKI will resolve with volume expansion. The most difficult point is to differentiate between HRS-AKI and intrarenal AKI. The patient meets the HRS criteria if there is no creatinine improvement after 48 h of volume expansion, and data do not suggest the presence of parenchymal disease such as proteinuria > 500 mg/day and/or microhematuria > 50 RBC per high power field. When HRS is suspected, prompt pharmacologic therapy with terlipressin or noradrenaline should be started [108]. If terlipressin is necessary, a continuous infusion is more effective and has fewer adverse effects than bolus administration, and the dose should be adjusted taking into account that the treatment goal is sCr of 1.5 mg/dL or less with a reduction of at least 50% [109].

**Recommendation.** Early diagnosis of AKI, avoidance non-steroidal anti-inflammatory drugs and metamizole, and rapid volume expansion with albumin are primary interventions that can improve outcomes. Differentiating HRS-AKI from non-HRS-AKI is essential as the treatments vary.

Level of evidence: B; Recommendation: Strong; Coincidence: 11/11.

#### 3.5.2. Encephalopathy


**Q18. How should we treat hepatic encephalopathy after surgery?**


HE is a clinical diagnosis that can be difficult to establish. Ammonia levels do not correlate with the clinical severity of HE; however, a normal ammonia level has a negative predictive value of 80%, suggesting an alternative cause of mental status changes. Blood samples should be drawn without a tourniquet and immediately sent to the laboratory. Monitoring ammonia levels as a response to therapy is not recommended because ammonia levels are unlikely to normalize and often will remain elevated after the resolution of HE [11,110].

HE should be controlled and reversed before non-emergent surgery by using lactulose and/or rifaximin. When HE is diagnosed, triggers of encephalopathy, including GI, bleeding, infection, central nervous system depressing medications, electrolyte disturbances, hypoxia, constipation, and renal insufficiency, should be ruled out [5]. In addition, if after surgery a patient has HE and is unable to take food by mouth, enemas of 300 mL lactulose in 1000 mL water can be given up to every 2 h. Bowel movements should be monitored; a nasogastric tube can facilitate the dosing of treatment if the patient cannot swallow. Lactulose 30–45 mL 3 to 4 times per day is generally required to achieve 2–4 soft stools per day. Rifaximin 550 mg twice daily can be added if lactulose is insufficient. Rifaximin can also be given for primary prophylaxis of HE after gastrointestinal hemorrhage [10,11].

**Recommendation.** Ammonia levels can be obtained to exclude or implicate HE as an etiology of altered mental status but not to follow its progression or response to therapy. After surgery, if a patient has HE and is unable to take food by mouth, lactulose enemas can be given. Bowel movements should be monitored, and oral lactulose and rifaximin can be added orally or through a nasogastric tube.

Level of evidence: A; Recommendation: Strong; Coincidence: 11/11.

#### 3.5.3. Ascites


**Q19. How should we manage ascites after surgery?**


Patients with ascites are at risk of pulmonary aspiration during induction of anesthesia, which can restrict pulmonary function and delay perioperative recovery. Therefore, in patients with clinical ascites, large-volume abdominal paracentesis should be performed preoperatively, with intravenous administration of albumin in a dose of 8 g for each liter of ascitic fluid removed [9].

In patients with CSPH, the administration of intravenous fluid and blood products should be limited perioperatively to avoid increasing extracellular volume, ascites, and increased risk of bleeding [111]. Patients with abdominal incisions may require therapeutic paracentesis more often or placement of an intra-abdominal drain to allow for controlled drainage of ascites and reduce the risk of abdominal wound dehiscence and abdominal wall herniation, as well as to avoid respiratory compromise [10]. Renal function and volume status should be monitored daily and promote judicious fluid and electrolyte management to avoid the accumulation of ascites or edema while maintaining intravascular volume to perfuse the kidneys [10].

Diagnostic paracentesis should be performed promptly if fever, abdominal pain, or new hepatic decompensation to rule out bacterial peritonitis. Usual primary prophylaxis and secondary prophylaxis for SBP should be followed in those with a low ascitic total protein concentration or history of SBP, as in any other patient with ACLD. After surgery, in patients who are not able to take oral medications, third-generation cephalosporins such as ceftriaxone in a dose of 1 g every 24 h intravenously should be administered. Quinolones have been considered an alternative of choice, but, currently, the rate of resistance to these antibiotics is high [112]. In fact, in patients who recently received treatment with quinolones, this family of antibiotics should be avoided due to the risk of the presence of resistant bacteria. Likewise, a carbapenem could be considered as an alternative in those patients in whom extended-spectrum beta-lactamases producing gram-negative bacilli were recently isolated.

If the patient is stable, a moderate restriction in sodium in the diet (80–120 mmol/day, corresponding to 4.6–6.9 g of salt) is recommendable. Once oral medications are tolerated, patients on chronic diuretics can restart them according to the previous dose and the actual volume status according to clinical guidelines, and norfloxacin prophylaxis can be resumed [36]. If diuretics are initiated, careful monitoring of creatinine and electrolyte levels is recommended. Large-volume paracentesis with albumin could be necessary for uncontrolled ascites or renal insufficiency [9].

**Recommendation.** Patients with abdominal incisions may require therapeutic paracentesis or intra-abdominal drain to reduce the risk of abdominal wound dehiscence, abdominal wall herniation, and respiratory compromise. Third-generation cephalosporins such as ceftriaxone can be administered prophylactically. If diuretics are initiated, careful monitoring of creatinine and electrolyte levels is recommended.

Level of evidence: B; Recommendation: Strong; Coincidence: 11/11.

#### 3.5.4. Acute-on-Chronic Liver Failure


**Q20. Can ACLD patients develop Acute-on-Chronic Liver Failure after surgery?**


ACLF is a recently recognized syndrome characterized by acute decompensation of cirrhosis and organ/system failure(s) such as the liver, kidney, brain, coagulation, circulation, and/or respiration with extremely poor survival (28-day mortality rate 30–40%). Potential precipitating events of ACLF are bacterial infections, gastrointestinal hemorrhage, active alcoholism, transjugular intrahepatic portosystemic shunting, therapeutic paracentesis without the use of intravenous albumin, hepatitis, and major surgery [113]. The general management of ACLF includes a rapid identification using the CLIF Consortium ACLF score (CLIF-C ACLFs) based on the age, white cell count, and the function of six organs (liver, kidney, brain, coagulation, circulation, and respiration) ranging from 0 to 100. Patients with ACLF should be treated in an ICU environment including measures that prevent the progression of the syndrome and the use of specific organ support systems. Extracorporeal liver support systems based on albumin dialysis were widely used. However, they did not improve survival, and potential candidates for liver transplantation should be transferred to a transplant center [113].

Recently, Klein et al. have described that the development of ACLF after surgery is frequent, especially in those ACLD patients with active bacterial infection, lower serum sodium, and kidney or coagulation dysfunction. Patients who developed ACLF within 28 days after surgery had higher mortality, and survival did not differ from those with ACLF at the surgery. Independent predictors of 1-year all-cause mortality were alkaline phosphatase, the MELD score, and preoperative HE. So, the authors concluded that patients with ACLD should be carefully managed perioperatively [114].

**Recommendation.** Involve a proficient multidisciplinary team with experience in early detection and treatment of ACLD decompensation and ACLF in the postoperative setting is recommended, as it can avoid progressive complications.

Level of evidence: B; Recommendation: Strong; Coincidence: 11/11.

## 4. Discussion

Compared with previously published reviews, our revision provides a structured global vision of the literature that supports the answers to 20 common questions in routine clinical practice asked by expert physicians. Patients with compensated ACLD and few comorbidities tolerate surgery well. Laparoscopic and robotic approaches are good options for liver resections in advanced chronic liver disease patients, but more studies are necessary. Elective cholecystectomy and abdominal hernia surgery must be avoided in decompensated patients. Liver transplantation or alternatives to surgery should be considered in high-risk patients (CTP class C or MELD >20) with a postoperative 3-month mortality rate greater than 15%. Viscoelastic testing allows a more accurate assessment of coagulation, and its use should be considered to take the transfusion decision with a restrictive strategy. If possible, regional anesthesia should be the choice. Nutrition and micronutrient supplementation should be started early after surgery, preferably using the enteral route. Opioids can be used safely, and the administration of NSAIDs for pain control should be avoided. Early diagnosis of infections and acute kidney injury are primary interventions that can improve outcomes.

### 4.1. Limitations

The limitation of our document is that the literature, despite being up-to-date and relevant, was not obtained through a systematic review of the databases.

### 4.2. Future Directions

The review makes it possible to find those questions whose answers are answered based on data published in lower-quality studies and opens the door to developing new studies and trials to improve the quality of the evidence obtained to date.

## 5. Conclusions

Our article is a review of the previously existing literature ordered and synthesized. It provides a conclusion for each of the 20 questions asked about clinical aspects. Authors consider that this review can affect public policies, simplifying, structuring, and helping the decision-making in the management of patients with advanced chronic liver disease undergoing surgery.

Recognition of patients with advanced chronic liver disease before an elective or emergency surgery is essential. A multidisciplinary approach is critical for optimizing all phases of perioperative care, including planning, risk evaluation, intraoperative management, and postoperative recovery.

## Figures and Tables

**Figure 1 life-13-00132-f001:**
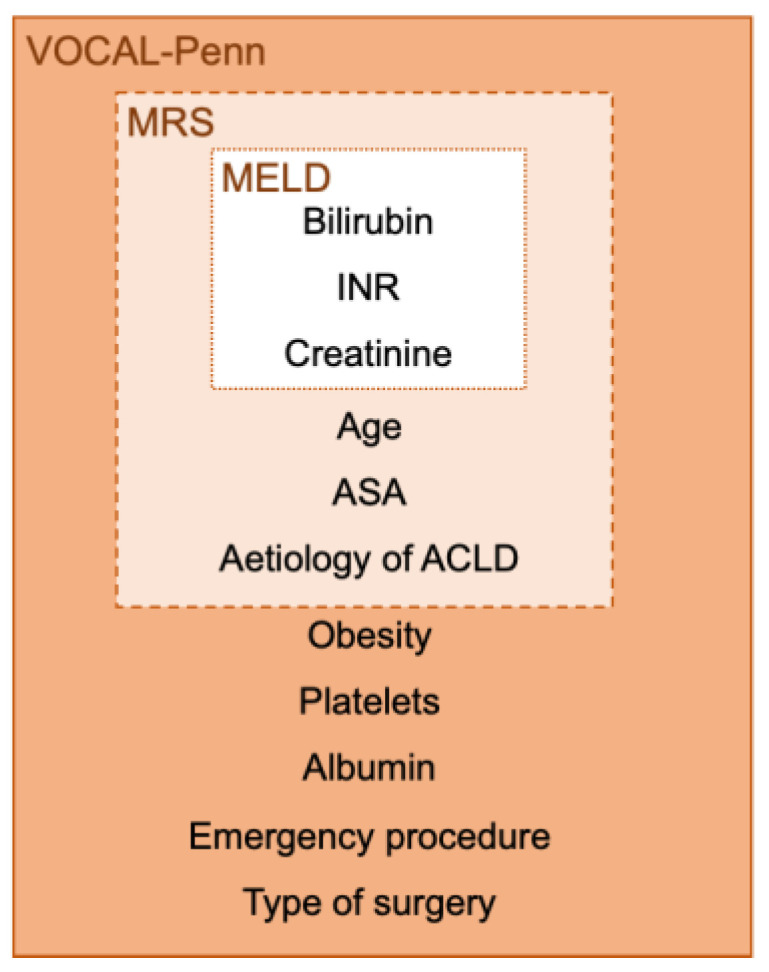
Surgical mortality risk in patients with advanced chronic liver disease. Main available tools for the assessment of surgical mortality risk before surgery. ACLD, Advanced chronic liver disease. INR, international normalized ratio. MELD, Model for End-Stage Liver Disease. MRS, Post-operative Mayo Risk Score. VOCAL, Veterans Outcomes and Costs Associated with the Liver.

**Figure 2 life-13-00132-f002:**
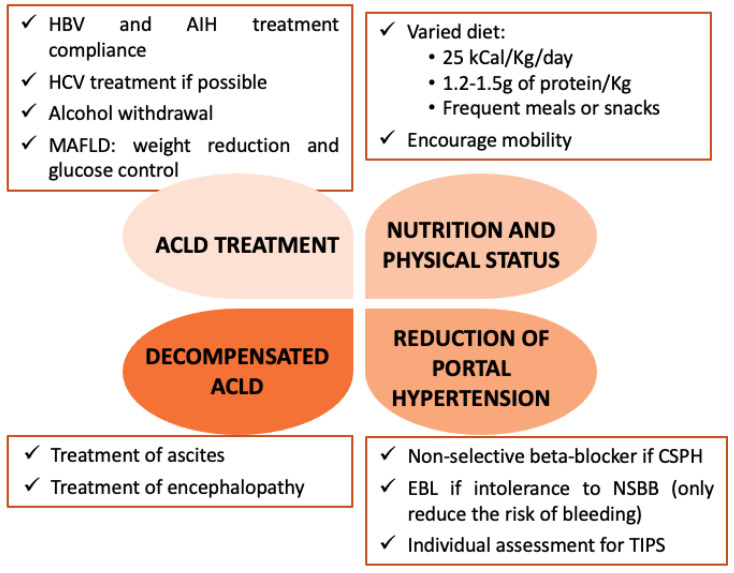
Preoperative management in patients with advanced chronic liver disease. Summary of the most important aspects of preoperative management to reduce surgical risk. ACLD, Advanced chronic liver disease. AIH, autoimmune hepatitis. CSPH, clinically significant portal hypertension. EBL, endoscopic band ligation. HBV, hepatitis B virus. HCV, hepatitis C virus. MAFLD, metabolic-associated fatty liver disease. NSBB, non-selective beta-blocker.

**Figure 3 life-13-00132-f003:**
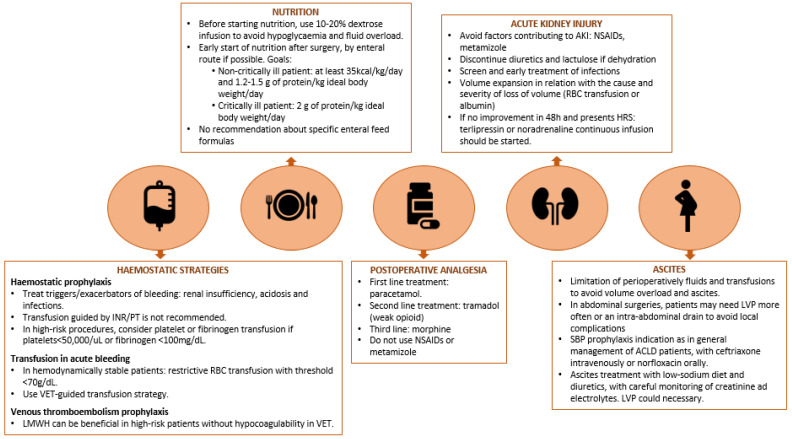
Postoperative management in patients with advanced chronic liver disease. Practical recommendations regarding hemostatic strategies, nutrition, analgesia, acute renal failure, and ascites. ACLD, Advanced chronic liver disease. AKI, Acute Kidney Injury. HRS, hepatorenal syndrome. INR, international normalized ratio. LMWH, low-molecular-weight heparin. LVP, large volume paracentesis. NSAIDs, nonsteroidal anti-inflammatory drugs. PT, prothrombin time. RBC, red blood cell. SBP, spontaneous bacterial peritonitis. VET, viscoelastic test.

**Figure 4 life-13-00132-f004:**
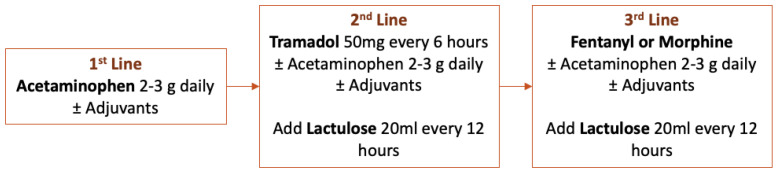
Algorithm of recommended analgesia in patients with advanced chronic liver disease. NSAIDs were not included in any of the three steps of treatment.

**Table 1 life-13-00132-t001:** Panel of key questions.

**Section 1. Preoperative evaluation and management**	
1.	What pathophysiological characteristics do patients with ACLD ^1^ present?
2.	How could we diagnose patients with ACLD ^1^ before surgery?
3.	What tools are available to assess the risk of surgical mortality before surgery?
4.	How could we intervene to reduce surgical risk before elective procedures?
5.	How can we evaluate and manage coagulation disorders?
**Section 2. Surgical procedures**	
6.	What recommendations should be considered for surgical procedures?
7.	What considerations should be taken into account for elective surgery?
8.	What concerns should be considered for emergent surgery?
**Section 3. Intraoperative management**	
9.	Is the anesthetic technique different?
10.	Is regional anesthesia recommended?
11.	Is cardiovascular monitoring necessary during surgery?
12.	What is the best fluid replacement strategy and vasopressor during surgery?
**Section 4. General recommendations in postoperative**	
13.	Is a liberal or restrictive transfusion strategy recommended in acute bleeding?
14.	Is venous thromboembolism prophylaxis recommendable after surgery?
15.	What is the treatment of postoperative pain?
16.	What nutritional recommendations should be made after surgery?
**Section 5. Postoperative management of hepatic decompensation and ACLD complications**	
17.	How should we manage an acute kidney injury after surgery?
18.	How should we treat hepatic encephalopathy after surgery?
19.	How should we manage ascites after surgery?
20.	Can ACLD ^1^ patients develop Acute-on-Chronic Liver Failure after surgery?

^1^ ACLD: Advanced chronic liver disease.

**Table 2 life-13-00132-t002:** Pros and cons of anesthetic drugs in advanced chronic liver disease.

Anaesthetic Drug	Pros	Cons
**Sevoflurane**	Fewest complications	
**Isoflurane**	Halogenated drug with lower hepatic metabolism. Little disturbance of hepatic blood flow.	
**Desflurane**	Little disturbance of hepatic blood flow.	
**Propofol**		Wake-up time slightly longer in ACLD patients due to their added extrahepatic metabolism.
**Diazepam, Clonazepam, Midazolam**		Prolonged effect in ACLD patients.
**Lorazepam,** **Oxacepam,** **Temazepam**	Minimally affected by liver disease.	
**Succinylcholine**		Longer elimination half-life in ACLD patients.
**Vecuronium**		Longer elimination half-life in ACLD patients.
**Rocuronium**		Longer elimination half-life in ACLD patients.
**Atracurium,** **Cis-atracurium, Mivacurium**	Metabolization without liver involvement.	
**Fentanyl,** **Sufentanil,** **Remifentanil**	Metabolism not affected in ACLD patients.	Can cause HE.

## Data Availability

This is a review of previously published literature, and no new data was created.

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
