# Peer review of "Clinical Guideline on Perioperative Management of Patients with Advanced Chronic Liver Disease"

_life, 2023, doi:10.3390/life13010132_

Round 1
Reviewer 1 Report
This is an extensive narrative review and consensus report on management of patients with advanced chronic liver disease undergoing hepatic and non-hepatic surgery.
Excellent work that covered all peri-operative aspects in these critical patients.
Main comment:
One of the aspect that would need further development is the risk of hepatic failure after liver surgery (which is increased mainly after major hepatectomy up to 8-12%). The preventive effect has been detailed in patient selection prior to intervention section but not the incidence according to type of surgery and the management.
Reviewer 2 Report
Dear authors
I had the pleasure to read your clinical guideline about perioperative management of patients with advanced chronic liver disease. I believe the work is sound. However, I have some minor comments and suggestions for the authors:
1: Lines 36-37. I find it odd that the authors describe the risk of HCC development in the first sentence. That is not the most essential information for the work. It is relevant to mention, but I recommend to do so later in the introduction.
2: Lines 44-46. Regarding "hepatic decompensation". I agree that ascites, HE, and variceal bleeding are events of cirrhosis decompensation. However, bacterial infections as well as development of ACLF, are complications to cirrhosis.
3: Materials and methods section. I suggest that the authors use a bit more efforts explaining why we need this guideline when guidelines already exist from the large liver associations. E.g. the EASL recommendation has not been updated since 2012, or that you apply a new and different and better approach to the create the guideline.
4: Table 1 - The abbreviation ACLD is incorrect.
5: Results - Pathophysiology: The authors provide a nicely written section about cirrhosis pathophysiology. In my opinion, I believe it is a bit off topic to dig that deep and suggest to shorten this section.
6: Figure 1a - When 1a and 1b are seperated, they should instead be named 1 and 2. Please, also present abbreviations in the figure legends.
7: Results - surgical risk prediction. This is a nicely written section, but again it is very long. As the VOCAL-Penn score has the best discriminative ability, I suggest to keep that section largely unchanged but shorten the previous section presenting the MRS.
8: Figure legend to Figure 1b and Figure 2 - please correct that it is not about acute on chronic liver disease, but advanced chronic liver disease. ACLF is something different.
9: You shortly mention quality of life. However, I suggest that the authors include more information on this. Patient-reported outcomes and quality-of-life measures have gained recognition the past years and have even succcesfully been incorporated in prognostic models/scores for patients with cirrhosis. As patient-reported outcomes reflect the patients view, they may also be important in the context of perioperative management. Actually, I do not believe much is written about this and you therefore have the opportunity to provide your expert view on the importance of PROs in this setting.
10. Line 351 - Please change to "MELD-Na", which is more widely used.
11. Section 3 - anaestetic techniques. This section is well-written. However, I would appreciate a table that summarizes the pros and cons for the drugs presented in the text.
12. Line 810. Guidelines for the management of ascites recommend a moderate sodium diet (80-120 mmol pr. day).
13. There is an error in the title for section 5. ACLF is not a decompensation but a syndrome that includes both acute decompensation and organ failures. I suggest changing the title to "Postoperative management of hepatic decompensation and cirrhosis complications".
I wish the authors best of luck with this and future work.
